# Forgery Detection in Digital Images by Multi-Scale Noise Estimation

**DOI:** 10.3390/jimaging7070119

**Published:** 2021-07-17

**Authors:** Marina Gardella, Pablo Musé, Jean-Michel Morel, Miguel Colom

**Affiliations:** 1Centre Borelli, ENS Paris-Saclay, Université Paris-Saclay, CNRS, 91190 Gif-sur-Yvette, France; jean-michel.morel@ens-paris-saclay.fr; 2IIE, Facultad de Ingeniería, Universidad de la República, Montevideo 11300, Uruguay; pmuse@fing.edu.uy

**Keywords:** blind estimation, forged image detection, heatmap, JPEG, noise level function

## Abstract

A complex processing chain is applied from the moment a raw image is acquired until the final image is obtained. This process transforms the originally Poisson-distributed noise into a complex noise model. Noise inconsistency analysis is a rich source for forgery detection, as forged regions have likely undergone a different processing pipeline or out-camera processing. We propose a multi-scale approach, which is shown to be suitable for analyzing the highly correlated noise present in JPEG-compressed images. We estimate a noise curve for each image block, in each color channel and at each scale. We then compare each noise curve to its corresponding noise curve obtained from the whole image by counting the percentage of bins of the local noise curve that are below the global one. This procedure yields crucial detection cues since many forgeries create a local noise deficit. Our method is shown to be competitive with the state of the art. It outperforms all other methods when evaluated using the MCC score, or on forged regions large enough and for colorization attacks, regardless of the evaluation metric.

## 1. Introduction

An escalating number of falsified images are being shared on the web and feeding fake news. Indeed, the popularization of digital devices as well as the development of user-friendly manipulation software have resulted in an increase in the traffic of manipulated content. The credibility of images is under question, and therefore, methods relying on scientific evidence are required to assess the authenticity of images.

Two different approaches have emerged to address this issue. On the one hand, techniques such as digital image watermarking prevent image forgery by embedding data at the moment of digitization. Such data can be detected or extracted later to authenticate the image [1]. Although these methods provide reliable authentication, they are limited to specifically equipped cameras.

On the other hand, passive methods that do not depend on prior knowledge have also been developed. These methods rely on the fact that image forgery techniques leave specific traces that can be detected as local inconsistencies in the image statistics [2,3]. Most classic methods aim to detect specific cues such as misalignment of the Bayer pattern or perturbations in the demosaicing traces [4,5,6], differences in the camera response function [7,8], or inconsistencies in the JPEG-compression grid or quality [9,10,11,12].

Recent deep-learning models have been developed to tackle the task of forgery detection [13]. These methods can be trained to detect specific falsification techniques such as splicing [14,15], copy-move [16,17] and inpainting [18,19], or to detect general attacks [20,21,22]. The main challenge shared by these methods is the construction of adequate training datasets ensuring good results on new real-world examples.

As first suggested by [3], noise residuals can provide substantial cues for detecting forgeries. Indeed, the initial Poisson noise [23] is transformed by multiple operations specific to each image formation process [24], leading to the final JPEG image. Hence, detecting noise inconsistencies is a rich source of forgery evidence. The use of noise residuals has evolved over time. Early methods [25,26] directly search inconsistencies in this residual whereas more recent algorithms use it as an input for further feature extraction [27,28]. Accurately estimating the residual noise traces after the complex set of transformations of the camera’s processing chain is the main challenge of this class of algorithms.

With these considerations in mind, we propose a noise-based method built on non-parametric multi-scale noise estimation [29]. The multi-scale approach has been shown to effectively deal with the correlations introduced by the demosaicing and JPEG-compression processes [30] and stands out as a suitable framework for noise inconsistency analysis.

The rest of the article is organized as follows. Section 2 reviews the image forgery detection techniques based on noise inspection. The proposed method is described in Section 3. Section 4 presents experimental results in addition to a comparison with other state-of-the-art techniques. The main conclusions are summarized in Section 5, where future work directions are also highlighted.

## 2. Related Work

The residual noise observable in images depends on the in-camera processing pipeline. It can therefore reveal the presence of tampered regions by detecting local inconsistencies in the noise statistics that are incompatible with a unique camera processing chain. Such inconsistencies can be produced by the forgery or its post-processing.

The most outstanding source of non-uniform noise is the photo-response non-uniformity (PRNU) which is caused by small differences in the way sensors react to the light source. PRNU-based forensics methods, such as [31,32,33], are mostly used for source camera identification. However, since PRNU varies across the image itself, it can also provide evidence of a local manipulation. The main limitation is that PRNU-based detection methods require access to a certain number of (untampered) images taken with the same camera, to accurately estimate the PRNU pattern.

Blind noise-based detection methods usually estimate noise variance locally to detect suspicious regions and then apply a classification criterion to locate forgeries. In [25], the noise variance is estimated in blocks using a median absolute deviation (MAD) estimator in the wavelet domain. Classification is performed using homogeneous noise standard deviation criteria. In turn, Ke et al. [34] proposes noise level estimation using principal component analysis (PCA) [35]. K-means is then applied to group image blocks into two clusters. A similar approach can be found in [36]. A different method was introduced in [37], where block-wise noise estimation is based on the observation that the kurtosis values across different band-passed filter channels are constant [38]. The method concludes by segmenting the image into regions with significantly different noise variances by k-means. In [39], the image is segmented using the simple linear iterative clustering (SLIC) algorithm. Then, for each region, five filters are used to extract noise. The computed noise features are then used for classification, which is performed by energy-based graph cut.

The aforementioned methods estimate a single and constant noise level, namely an additive white Gaussian noise (AWGN) model. However, this hypothesis does not hold in realistic scenarios since noise levels depend on the image intensity [40]. More recent methods consider this fact and estimate a noise level function (NLF) rather than a single noise level. In [41], the authors proposed to jointly estimate the NLF and the camera response function (CRF) by segmenting the image into edge and non-edge regions. Noise level functions are then compared and an empirical threshold is fixed in order to detect salient curves. The methods introduced in [42,43] instead analyze a histogram based on the noise density function at the local level in order to reveal suspicious areas. The method proposed in [44] computes an NLF-based on Wiener filtering. Local noise levels in regions with a certain brightness are assumed to follow a Poisson distribution, according to which, the larger the distance to the NLF, the higher the probability of forgery. On the other hand, the approach developed in [45] consists of estimating a noise level function that depends on the local sharpness rather than on the intensity.

Recently, forgery detection methods based on deep learning and feature modeling have been developed. The method reported in [27] proposes using noise residuals to extract local features and compute their co-occurrence histograms, which are then classified in two classes using the expectation–maximization algorithm. More recently, the same authors presented a novel CNN-based method for noise residual extraction [28]. A similar approach can be found in [46]. On the other hand, Zhou et al. [47] proposed a two-stream CNN, one for the detection of tampering artifacts and the other to leverage noise features. Deep learning-based methods are more general than previously described ones. A major limitation of these methods is that they require large training datasets, which are not always available. Furthermore, their performance generally remains dataset dependent.

## 3. The Proposed Method

We propose a new method for JPEG-compressed image forgery detection based on multi-scale noise estimation. The method addresses the fact that, after going through the complete camera processing pipeline, noise is not only signal-dependent but also frequency-dependent. In particular, after demosaicing, noise becomes spatially correlated, and furthermore, the quantization of the DCT coefficients during JPEG-compression differently affects the noise at each frequency. In this context, multi-scale noise estimation is the most suitable approach since it enables capturing noise at medium and low frequencies.

Let *I* be an image with *C* color channels. We first split the image into W×W blocks with 1/2 overlap, extending the image in the borders by mirroring if necessary. We will refer to these blocks as macroblocks.

For each color channel, we estimate the global image noise curve as well as the local noise curves for each macroblock using an adaptation of the technique [29], described in Appendix A. For each channel, we compare the global noise curve with the ones locally obtained by computing the number of bins of the local noise curve that are below the global noise curve. By doing so, we obtained a heatmap for each channel that shows, for each macroblock, the percentage of bins in its noise curve whose count is below the global estimation. The information contained in the *C* obtained heatmaps is then combined by taking their geometric mean. As a result, we obtain a single heatmap.

For non-forged images, we expect the macroblocks to show similar noise levels functions as the one computed for the whole image. However, noise estimation is highly affected by image content. Indeed, noise overestimation is expected to happen in textured regions [48]. As a consequence, local noise curves computed over textured areas may be above the global one, even if no tampering has been performed. To prevent this kind of macroblock being perceived as suspicious, we only consider the number of bins below the global noise curve. Indeed, the global noise curve provides a lower bound for local noise curves since the noise estimation algorithm [29] has more samples from which to choose the adequate ones to estimate noise. Therefore, local noise curves that are below the global one are suspected to correspond to a different source. Figure 1 depicts the previously described situation. Indeed, we can observe that the non-forged macroblock shows higher noise levels than the global image, even though it is not tampered. On the other hand, the manipulated macroblock exhibits lower noise levels.

The next step consists of repeating the previously described process but replacing the image *I* and the macroblocks by their down-scaled version. To this aim, let *S* be the operator that tessellates the image into sets of 2×2 pixels blocks, and replaces each block by the average of the four pixels. We define $S_{n}\left(I\right)$ as the *n*-th scale of an image *I* obtained by applying *n* times the operator *S* to the image *I*. This procedure allows noise curves to show the noise contained in lower frequencies and can provide further evidence of tampering that could be hidden under strong JPEG-compression.

By iterating the process at successive scales, we obtain one heatmap per scale which shows the geometric mean of the percentages obtained at each channel. Each of these heatmaps may provide useful information to detect tampering since they account for noise contained at different frequencies. The sum of the heatmaps obtained at the different scales is computed and then normalized in the [0,255] interval. To obtain the final heatmap, for each pixel we compute the average of the values of each macroblock containing it.

The residual noise present in images having undergone demosaicing and JPEG-compression is correlated and therefore creates medium-sized noise spots. This may cause the blocks of size 8×8 used for noise estimation to fit inside these spots, thus causing noise underestimation. Again, estimating noise in sub-sampled versions of the image enables these spots to fit inside the scanning blocks and to accurately measure low-frequency noise. We propose repeating the sub-scaling process until reaching $S_{2}\left(I\right)$, as suggested in [30].

Further scales could be also considered. However, the most relevant information is already retrieved at $S_{2}$. Furthermore, the macroblock’s size would become critically small and unfit to estimate noise curves: if the original macroblocks are sized W×W in $S_{0}$, in $S_{1}$ they will be of size (W/2)×(W/2), and in $S_{2}$ of size (W/4)×(W/4). Indeed, as shown in Appendix B, the best performance for the proposed method is achieved when considering macroblocks of size W=256. In this context, the macroblocks are sized 128×128 in $S_{1}$ and 64×64 in $S_{2}$.

Figure 2 shows the pipeline of the proposed method, from the moment that the algorithm is fed with the input image until the final heatmap is delivered. Additionally, a summarized version of the proposed method is given below.

Given a suspect image and the parameters for the method (macroblock side, stride and number of scales), the proposed algorithm goes as follows:Open the suspect image.Get a list of all macroblocks according to the given macroblock size and the considered stride.For each scale and each color channel, estimate the global NLF of the image and compare it to NLF computed at each macroblock. We are interested in the percentage of histogram bins below the global curve.To obtain the final result of the algorithm, the heatmaps obtained at each of the scales are combined.

Please refer to Algorithm 1 for a detailed pseudo-code description. The actual source code is available at (accessed on 31 May 2021) https://github.com/marigardella/PB_Forgery_Detection, together with the instructions and requirements to run the method. Further implementation details are given in Appendix C.
**Algorithm 1** Pseudo-code for the proposed method**Input:** image *I* of shape Nx×Ny with *C* color channels.**Parameters:**
W=256 macroblock side, S=0.5 stride, num_scales = 3 number of scales.
1:$M_{x}=\left\lfloor N_{x}/\left(W\times S\right)\right\rfloor -1.$         ▹ horizontal number of macroblocks2:$M_{y}=\left\lfloor N_{y}/\left(W\times S\right)\right\rfloor -1.$            ▹ vertical number of macroblocks3:macroblocks_list←listofallW×WmacroblockswithSstride.4:**for** each scale *s* **do**5:    **for** each channel *c* **do**6:        $I_{s}^{c}\leftarrow \;\mathrm{get}\;\mathrm{image}\;\mathrm{in}\;\mathrm{scale}\;s\;\mathrm{and}\;\mathrm{channel}c.$7:        $f_{I_{s}^{c}}\leftarrow \;\mathrm{noise}\;\mathrm{curve}\;\mathrm{estimation}\;\mathrm{for}\;I_{s}^{c}\;\mathrm{using}$ [29] as described in Appendix A.8:        $H^{c}\leftarrow \;\mathrm{zeros}\left(M_{x}\times M_{y}\right).$9:        **for** each macroblock in macroblocks_list **do**10:           $M_{s}^{c}\leftarrow \;\mathrm{get}\;\mathrm{macroblock}\;\mathrm{in}\;\mathrm{scale}\;s\;\mathrm{and}\;\mathrm{channel}\;c.$11:           $f_{M_{s}^{c}}\leftarrow \;\mathrm{noise}\;\mathrm{curve}\;\mathrm{estimation}\;\mathrm{for}\;M_{s}^{c}\;\mathrm{using}$ [29] as described in Appendix A.12:           $H^{c}\left[M_{s}^{c}\right]\leftarrow \;\mathrm{percentage}\;\mathrm{of}\;\mathrm{bins}\;\mathrm{of}\;f_{M_{s}^{c}}\;\mathrm{below}\;f_{I_{s}^{c}}.$13:        **end for**14:    **end for**15:    $H_{s}\leftarrow \;\mathrm{geometric}\;\mathrm{mean}\;\mathrm{of}\;\mathrm{the}\;\mathrm{heatmaps}\;H^{c}.$16:**end for**17:$H_{\mathrm{aux}}\leftarrow \;\mathrm{sum}\;\mathrm{and}\;\mathrm{normalization}\;\mathrm{of}\;\mathrm{heatmaps}\;H_{s}.$18:$H\leftarrow \;\mathrm{compute}\;\mathrm{for}\;\mathrm{each}\;\mathrm{pixel}\;\mathrm{the}\;\mathrm{average}\;\mathrm{of}\;H_{\mathrm{aux}}\;\mathrm{for}\;\mathrm{each}\;\mathrm{macroblock}\;\mathrm{containing}\;\mathrm{it}.$19:**return***H*.


## 4. Experimental Results

We conducted two experiments. First, we evaluated the relevance of the multi-scale approach by comparing the results obtained using a single scale ($S_{0}\left(I\right)$), two sub-scales ($S_{0}\left(I\right)$ and $S_{1}\left(I\right)$) and three sub-scales ($S_{0}\left(I\right)$, $S_{1}\left(I\right)$ and $S_{2}\left(I\right)$). Second, we compared our method with state-of-the-art forgery-detection algorithms based on noise analysis.

Datasets

All experiments were conducted on the CG-1050 database [49] which contains four datasets, each one corresponding to a different forgery technique: colorization, copy-move, splicing and retouching. The total number of forged images is 1050. This database is varied in nature, including images captured in 10 different places. The size of the images is 3456×4608 or 4608×3456 pixels. The database includes both RGB and grayscale images, all of which are JPEG-compressed. The estimated JPEG-quality [50] for each dataset is shown in Table 1.

Forgery masks were constructed by computing the absolute difference between the original image and the forged one in each channel. To avoid pixels whose values had changed due to global manipulations rather than tampering, the difference from one image to another was thresholded. Only pixels whose value varied more than this threshold for at least one channel were kept. Masks were then further refined in order to prevent isolated pixels from being regarded as forged. The thresholds used were 15 for the copy-move, colorization and splicing datasets and 10 for the retouching one.

The distribution of the mask’s size on each of the four datasets is shown in Figure 3.

Evaluation Measures

Forgery localization is a particular case of binary classification. Indeed, there are two possible classes for each pixel: forged (positive) or non-forged (negative). Performance measures are usually based on the confusion matrix [51], which has four values, each one corresponding to the four possible combinations of predicted and actual classes, as shown in Figure 4.

Three metrics based on these four quantities are proposed in order to compare the results obtained in both experiments. Namely, we evaluated the results using the IoU, the F1 and the MCC scores, defined as
$$\begin{array}{cc} \mathrm{MCC} & =\frac{\mathrm{TP}\times \mathrm{TN}-\mathrm{FP}\times \mathrm{FN}}{\sqrt{\left(\mathrm{TP}+\mathrm{FP})\times (\mathrm{TP}+\mathrm{FN})\times (\mathrm{TN}\;+\;\mathrm{FP})\times (\mathrm{TN}\;+\;\mathrm{FN}\right)}}\;,\\ \mathrm{IoU} & =\frac{\mathrm{TP}}{\mathrm{TP}\;+\mathrm{FN}\;+\;\mathrm{FP}}\;,\\ F1 & =\frac{2\mathrm{TP}}{2\mathrm{TP}\;+\mathrm{FN}\;+\mathrm{FP}}\;. \end{array}$$
where TP stands for true positive, TN for true negative, FN for false negative and FP for false positive.

These metrics are designed to evaluate binary-estimated masks. However, all of the methods analyzed in this paper propose continuous heatmaps rather than binary masks. To adapt the metrics to the continuous setting, we used their weighted version. In this approach, the value of a heatmap *H* at each pixel *x* is regarded as the probability of forgery of the pixel. Therefore, we define the weighted true positives, weighted true negatives, weighted false negatives and weighted false positives as:$$\begin{array}{cc} {\mathrm{TP}}_{w} & =\sum_{x}H\left(x\right)\times M\left(x\right),\\ {\mathrm{TN}}_{w} & =\sum_{x}\left(1-H\left(x\right)\right)\times \left(1-M\left(x\right)\right),\\ {\mathrm{FN}}_{w} & =\sum_{x}H\left(x\right)\times \left(1-M\left(x\right)\right),\\ {\mathrm{FP}}_{w} & =\sum_{x}\left(1-H\left(x\right)\right)\times M\left(x\right), \end{array}$$
respectively, where *H* is the output heatmap normalized between 0 and 1, and *M* is the ground-truth binary mask where pixels with a value of 1 are forged. Then, the weighted version of the IoU, F1 and MCC scores are obtained replacing TP, TN, FN and FP with their weighted versions. It is important to point out that for some of the methods, the output is a two-sided heatmap (meaning that suspicious regions can appear in lighter or darker colors). Taking this into consideration, both the output heatmap and the inverted one are evaluated and only the highest score is kept.

### 4.1. Relevance of the Multi-Scale Approach

We first examined the pertinence of a multi-scale scheme. For this purpose, we computed the results obtained when considering one single scale $S_{0}\left(I\right)$ (which would correspond to the input image), using two scales $S_{0}\left(I\right)$ and $S_{1}\left(I\right)$, and using three scales $S_{0}\left(I\right),S_{1}\left(I\right)$ and $S_{2}\left(I\right)$. The scores obtained for each of these settings are shown in Table 2.

We can observe that using multiple scales leads to better results compared to a single one. Indeed, in all four datasets, the scores obtained by PB2 and PB3 are better than those obtained by PB1 for the three metrics. Regarding the number of scales yielding a better performance, the use of three scales obtains the best scores for the retouching, colorization and splicing datasets, whereas the use of two scales achieves a better performance in the copy-move dataset. However, the results obtained for the copy-move dataset are poor for the three variants of the method, and furthermore, they have very similar scores. We conclude that the use of three scales, $S_{0}\left(I\right)$, $S_{1}\left(I\right)$ and $S_{2}\left(I\right)$, gives the best performance among the evaluated alternatives. In fact, given that JPEG-compression is applied in 8×8 blocks without overlap, it is at $S_{2}$ that the most accurate noise estimation is achieved since we are able to capture noise contained in lower frequencies, which is less affected by the quantization of the DCT coefficients.

### 4.2. Comparison with State-of-the-Art Methods

In order to assess the performance of our method, we compared the results obtained on the CG-1050 dataset with those delivered by state-of-the-art noise-based methods: Splicebuster [27], Noiseprint [28], Mahdian [25], Pan [26], Zeng [36], Zhu [45] and Median [52]. For each algorithm, we used a publicly available implementation [53]. Table 3 lists all the evaluated methods as well as their reference article and the link to the source code used for the comparison.

The obtained results are given in Table 4. We observe that Splicebuster outperforms the rest of the methods in the retouching and splicing datasets regardless of the metric.

Our method ranks first for colorization attacks for all the three metrics considered. This forgery technique shows the relevance of considering noise curves instead of single noise levels. Indeed, when changing the color in a region of the image, noise levels are not necessarily perturbed. However, those noise levels will not be consistent with the new intensity but with the original. Estimating noise curves as the proposed method does enables detecting this kind of inconsistency which only appears when considering intensity-dependent noise models.

Regarding the copy-move dataset, Splicebuster delivers the best results when considering the F1 and IoU scores. However, our approach obtains the best MCC score.

The average ranking shows that Splicebuster outperforms the rest of the methods when considering both the F1 and IoU scores, followed by our method. Nevertheless, our method achieves the best average ranking when considering the MCC score, followed by Splicebuster.

Noiseprint stands out as the third best performing method for the IoU and F1 scores. It even ranks second for retouching and copy-move attacks when considering these scores. However, it shows a poor performance for the colorization dataset. This can be explained by the fact that the camera signature is left unchanged when performing this kind of manipulation.

The Pan and Mahdian methods are middle-ranked, showing better results when considering the MCC score. Finally, Median, Zeng and Zhu show the worst performance of all the considered methods regardless of the metric considered.

All of the evaluated methods have different resolutions which may affect their performance when forgeries are too small. To analyze the effects of the size of the forgeries, we computed the average score as a function of the forgery size. Figure 5 shows the average score obtained by each method when setting different lower bounds for the forgery size in each of the datasets considered.

The results suggest that our method outperforms the state-of-the-art approaches when considering large forgeries in all the datasets regardless of the considered score. The fact that it does not perform that well when considering small manipulations is a direct consequence of the size of the macroblocks. Indeed, for our method to provide reliable detection, the tampered region should be at least of the size of one of the tested macroblocks. In contrast, the performance of Splicebuster decreases as we consider larger forgeries. This is partially expected since the Gaussian-uniform model used in this method is better suited for small forgeries, as suggested by their authors in the original paper [27].

For further evaluation, we used the visual inspection of the results obtained by the proposed method and state-of-the-art approaches. Figure 6 shows examples of the outputs obtained by these methods for the colorization and retouching attacks, respectively, as well as for the corresponding original untampered images.

For the colorization attack shown in Figure 6, we can observe that, for all of the approaches except ours, the heatmap obtained when applying the method to the forged and original images are very similar. None of these methods is able to distinguish the tampered region by detecting the traces of the forgery. Instead, the proposed method provides a significant difference between the forged and pristine image; we observe that the forgery clearly stands out while for the pristine image, the values of the heatmap in that area are moderated.

In the case of retouching, we observe that all of the methods point out the forged region or at least part of it as suspicious. However, several interpretation problems arise. When analyzing the results provided by Splicebuster, we can notice that the heatmap corresponding to the tampered image precisely points to the border of part of the forgery. However, when considering the pristine image, there are several areas of the heatmap showing the same values, even if they are not tampered. The Noiseprint results better localize the forgery even though false alarms are present in the pristine image. Mahdian, Pan, Median, Zeng and Zhu methods show a further drawback: in the heatmap corresponding to the manipulated image, the forged regions stand out at the same level as other non-tampered parts of the image. The interpretation of the heatmaps is left to the user who has to decide whether the regions detected as suspicious should be considered forged or discarded. On the other hand, our method is able to localize the forgery when applied to the tampered image while showing no extreme values for the pristine one, making it easier for users to interpret.

## 5. Conclusions, Limitations and Future Research

In the fight against disinformation, the use of objective methods able to detect manipulated multimedia content becomes crucial. Providing such tools is the aim of the digital forensics research community, and in particular, of the present work. We believe that image forgery detection is a key resource to fight fake news.

JPEG images are broadly used and clearly stand out as one of the most popular image formats. From the acquired raw image to the final JPEG format delivered by the camera, a complex processing chain is applied. Along this process, the originally Poisson-distributed noise undergoes several transformations, resulting in a complex noise structure in the JPEG image whose model does not match the AWGN hypothesis. Noise inconsistency analysis is a rich resource for forgery detection given that forged regions are likely to have undergone a different processing pipeline or an out-of-camera manipulation. However, noise-based methods require accurately dealing with the changes induced by the successive steps of the camera processing chain.

In the present paper, we proposed a method that can correctly deal with the complex noise residuals observable in the JPEG image. The proposed method implements a multi-scale approach which has shown to be suitable for analyzing the highly correlated noise present in JPEG-compressed images.

Our comparative results show that our method outperforms state-of-the-art approaches when evaluating the results with the MCC score. For colorization attacks, our method performs best, regardless of the metric. In addition, when the size of the forgeries is large enough, our method shows the best performance in all the datasets, for all three considered metrics.

Nevertheless, the proposed method has its own limitations, mainly related to too-small and too-large forgeries. Indeed, if the forgery is too small with respect to the macroblock’s size, the method is likely to miss it. On the other hand, if the forgery is comparatively too large, the global noise curve may be distorted by the tampered region. The method is also by construction unable to detect a pure internal copy-move. Indeed, such a manipulation leaves the noise model unaltered. As a final negative note, the method cannot detect splicing when the forged region has more noise than the background image.

Future work includes refining the noise estimation step to use smaller macroblocks and thus improving the localization capabilities of our method.

## Figures and Tables

**Figure 1 jimaging-07-00119-f001:**
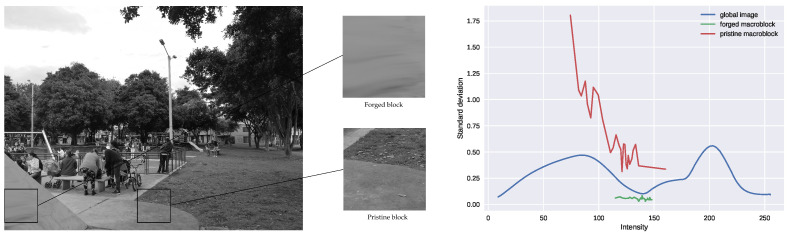
Estimated noise curves for the global image and for two macroblocks—one of which is contained in the manipulated region and the other is coming from the non-manipulated part of the image.

**Figure 2 jimaging-07-00119-f002:**
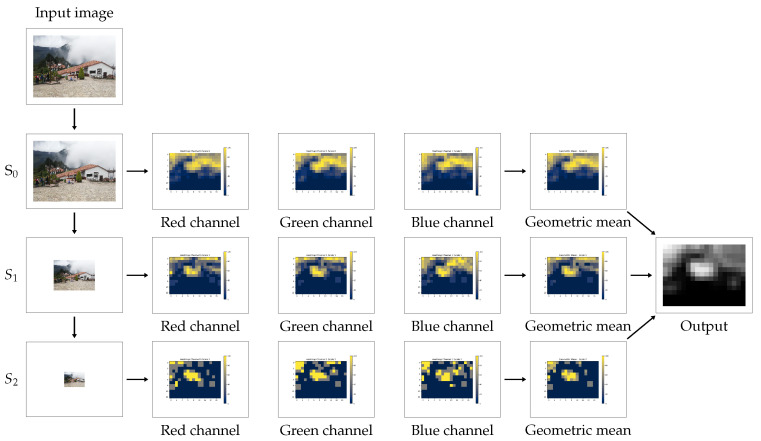
Complete pipeline of the method: successive scales are extracted from the input image. At each scale, one heatmap per color channel is computed and then combined according to their geometric mean. Finally, the obtained heatmaps at each scale are summed and normalized to produce the final output.

**Figure 3 jimaging-07-00119-f003:**
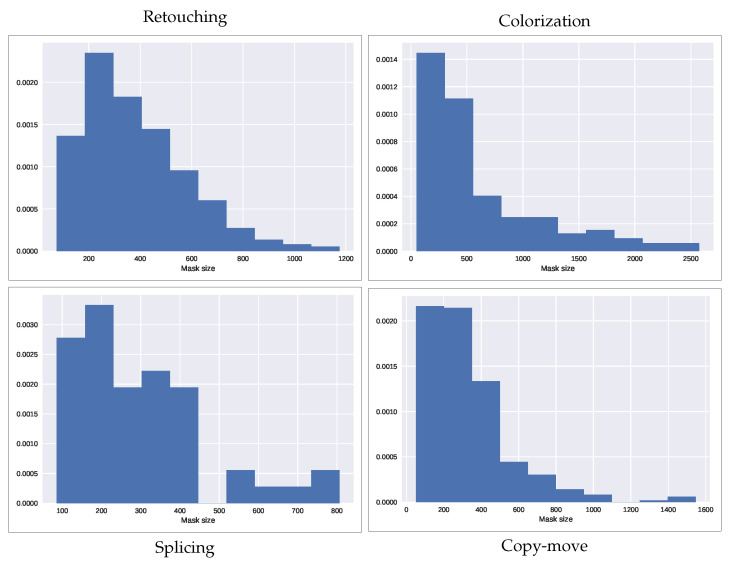
Distribution of the forgery size in each of the datasets considered. The forgery size is shown as the square root of the mask size, which represents the side of its equivalent square.

**Figure 4 jimaging-07-00119-f004:**
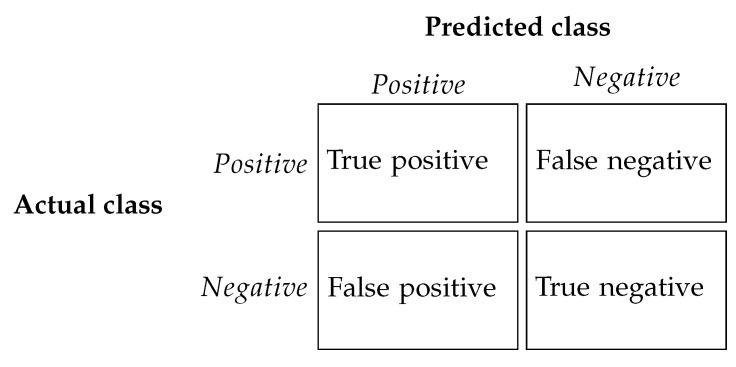
Confusion matrix: rows represent the actual classes while columns represent the prediction. The matrix has four possible values, corresponding to the four possible combinations of predicted and actual classes.

**Figure 5 jimaging-07-00119-f005:**
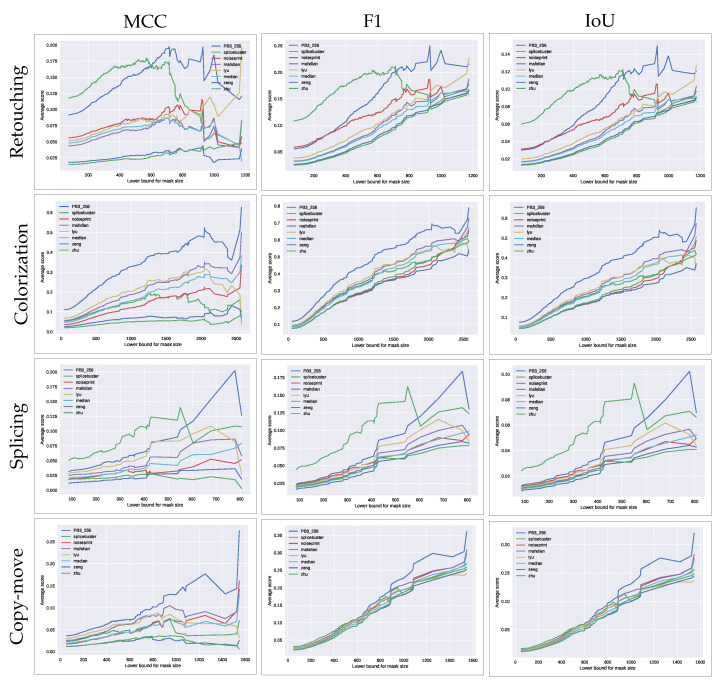
Average weighted MCC (**left**), IoU (**middle**) and F1 (**right**) scores obtained by each method as a function of the lower bound for the forgery size, in each of the datasets considered. Forgery size is shown as the square root of the mask size, which represents the side of its equivalent square.

**Figure 6 jimaging-07-00119-f006:**
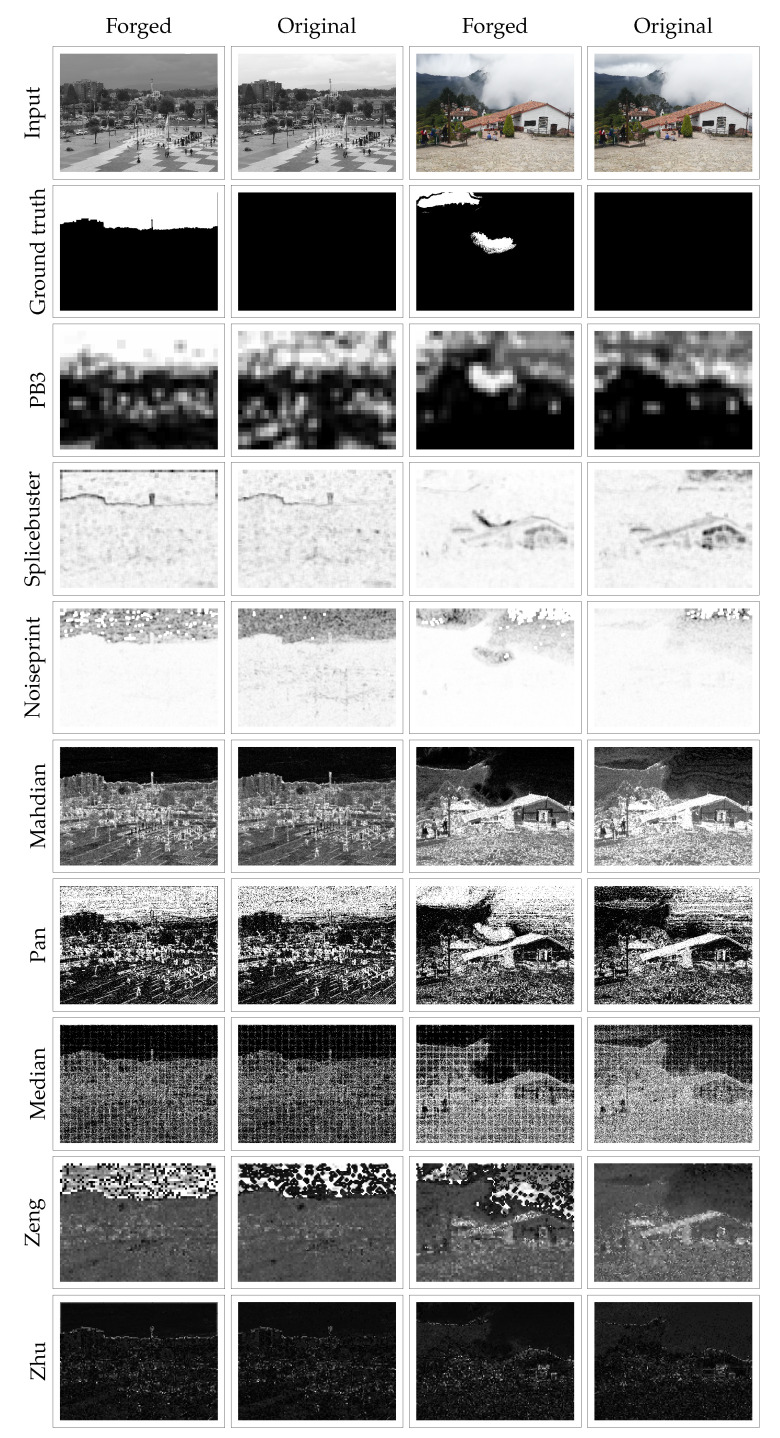
Results obtained for examples where colorization (**first column**) and retouching (**third column**) were performed, as well as for their corresponding original images (**second and fourth columns**). On the successive rows, the results obtained by each of the approaches for these images.

**Table 1 jimaging-07-00119-t001:** Average JPEG-quality and range for each of the datasets.

	Retouching	Colorization	Splicing	Copy-Move
Average JPEG-quality	86.9	86.8	87.3	86.8
JPEG-quality range	[71,88]	[71,88]	[71,88]	[71,88]

**Table 2 jimaging-07-00119-t002:** MCC, IoU and F1 scores for our method with one scale (PB1), two scales (PB2) and three scales (PB3).

**MCC**				
	**Retouching**	**Colorization**	**Splicing**	**Copy-Move**
PB1	0.0672	0.0958	0.0276	**0.0380**
PB2	0.0848	0.1066	0.0310	0.0377
PB3	**0.0915**	**0.1108**	**0.0316**	0.0362
**IoU**				
	**Retouching**	**Colorization**	**Splicing**	**Copy-Move**
PB1	0.0242	0.0721	0.0112	0.0148
PB2	0.0284	0.0756	0.0122	**0.0149**
PB3	**0.0300**	**0.0761**	**0.0123**	0.0145
**F1**				
	**Retouching**	**Colorization**	**Splicing**	**Copy-Move**
PB1	0.0454	0.1122	0.0216	0.0281
PB2	0.0529	0.1175	0.0234	**0.0282**
PB3	**0.0557**	**0.1192**	**0.0236**	0.0276

**Table 3 jimaging-07-00119-t003:** State-of-the-art methods used for the comparison as well as their reference and link to source code.

Method	Ref.	Source Code
Mahdian	[25]	https://github.com/MKLab-ITI/image-forensics (accessed on 31 May 2021)
Pan	[26]	https://github.com/MKLab-ITI/image-forensics (accessed on 31 May 2021)
Zeng	[36]	https://github.com/MKLab-ITI/image-forensics (accessed on 31 May 2021)
Median	[52]	https://github.com/MKLab-ITI/image-forensics (accessed on 31 May 2021)
Splicebuster	[27]	http://www.grip.unina.it/research/83-multimedia_forensics (accessed on 31 May 2021)
Noiseprint	[28]	http://www.grip.unina.it/research/83-multimedia_forensics (accessed on 31 May 2021)
Zhu	[45]	https://github.com/marigardella/Zhu_2018 (accessed on 31 May 2021)

**Table 4 jimaging-07-00119-t004:** Results of the evaluated methods measured by the average weighted IoU, F1 and MCC scores for each dataset that maximized the score.

**MCC**					
	**Retouching**	**Colorization**	**Splicing**	**Copy-Move**	**Average Ranking**
PB3	0.0915 (2)	**0.1108** (1)	0.0316 (2)	**0.0362** (1)	1.5
Splicebuster	**0.1176** (1)	0.0535 (4)	**0.0502** (1)	0.0233 (4)	2.5
Mahdian	0.0434 (6)	0.0566 (3)	0.0247 (4)	0.0257(3)	4
Pan	0.0513 (4)	0.0681 (2)	0.0282 (3)	0.0306 (2)	2.75
Noiseprint	0.0558 (3)	0.0361 (6)	0.0182 (6)	0.0177 (6)	5.25
Median	0.0479 (5)	0.0469 (5)	0.0204 (5)	0.0195 (5)	5
Zeng	0.0180 (7)	0.0262 (7)	0.0119 (8)	0.0117 (8)	7.5
Zhu	0.0147 (8)	0.0201 (8)	0.0180 (7)	0.0123 (7)	7.5
**IoU**					
	**Retouching**	**Colorization**	**Splicing**	**Copy-Move**	**Average Ranking**
PB3	0.0300 (3)	**0.0761** (1)	0.0123 (2)	0.0145 (2)	2
Splicebuster	**0.0600** (1)	0.0577 (2)	**0.0242** (1)	**0.0166** (1)	1.25
Mahdian	0.0168 (5)	0.0548 (4)	0.0102 (5)	0.0131(5)	4.75
Pan	0.0198 (4)	0.0576 (3)	0.0109 (4)	0.0138 (4)	3.75
Noiseprint	0.0312 (2)	0.0450 (7)	0.0114 (3)	0.0142 (2)	3.5
Median	0.0163 (6)	0.0513 (5)	0.0095 (7)	0.0123(6)	6
Zeng	0.0136 (7)	0.0441 (8)	0.0084 (8)	0.0114 (8)	7.75
Zhu	0.0129 (8)	0.0453 (6)	0.0102 (5)	0.0116(7)	6.5
**F1**					
	**Retouching**	**Colorization**	**Splicing**	**Copy-Move**	**Average Ranking**
PB3	0.0557 (3)	**0.1192** (1)	0.0236 (2)	0.0276 (2)	2
Splicebuster	**0.1081** (1)	0.0965 (2)	**0.0448** (1)	**0.0314** (1)	1.25
Mahdian	0.0324 (5)	0.0902 (4)	0.0199 (6)	0.0250(5)	5
Pan	0.0380 (4)	0.0946 (3)	0.0211 (4)	0.0264 (4)	3.75
Noiseprint	0.0588 (2)	0.0778 (7)	0.0222 (3)	0.0271 (3)	3.75
Median	0.0315 (6)	0.0857 (5)	0.0185 (7)	0.0236 (6)	6
Zeng	0.0264 (7)	0.0765 (8)	0.0165 (8)	0.0220 (8)	7.75
Zhu	0.0250 (8)	0.0779 (6)	0.0200 (5)	0.0224(7)	6.5

## Data Availability

Images from the CG-1050 database [49] were used in this article. The full database and documentation can be downloaded from https://data.mendeley.com/datasets/dk84bmnyw9/2 (accessed on 31 May 2021). The source codes for Splicebuster [27] and Noiseprint [28] methods are available at http://www.grip.unina.it/research/83-multimedia_forensics (accessed on 31 May 2021). The source codes for Pan [26], Mahdian [25], Median [52] and Zeng [36] algorithms are available at https://github.com/MKLab-ITI/image-forensics/blob/master/matlab_toolbox (accessed on 31 May 2021). The implementation of the Zhu [45] method is available at https://github.com/marigardella/Zhu_2018 (accessed on 31 May 2021). Finally, the source code for the proposed method is available at https://github.com/marigardella/PB_Forgery_Detection (accessed on 31 May 2021).

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
