# Peer review of "Forgery Detection in Digital Images by Multi-Scale Noise Estimation"

_2313-433X, 2021, doi:10.3390/jimaging7070119_

Round 1
Reviewer 1 Report
The paper proposes a noise-based method multi-scale noise estimation. The approach which has shown to be suitable to analyze the highly correlated noise present in JPEG-compressed images. It has certain workload, but still needs to modify more perfect. My suggestions are as follows:
- The main algorithm idea is not explained clearly in abstract, and background is not clear;
- Material and structure of the paper were no innovation. This manuscript does not bring any new knowledge or data on materials property;
- No important references in the field are cited, and the research content is too old and not innovative. Do not contribute anything new;
- Analysis of research results is not in-depth. The analysis overemphasized the importance of the results and did not explain the value of the research in depth.
Author Response
We would like to thank you for taking the time to assess our manuscript. We address your suggestions below and in the updated version of the manuscript. Major changes in the manuscript are highlighted in red.
- The main algorithm idea is not explained clearly in abstract, and background is not clear;
We updated the abstract by including some further explanation on the main idea behind our method.
- Material and structure of the paper were no innovation. This manuscript does not bring any new knowledge or data on materials property;
The proposed method is new and well-performing. Furthermore, it is the first method to address the highly correlated noise present in JPEG images by introducing a multi-scale approach. None of the already existing noise-based methods for forgery detection do this.
- No important references in the field are cited, and the research content is too old and not innovative. Do not contribute anything new;
We have updated the references in order to include more deep-learning approaches. As mentioned before, up to our knowledge, this is the first method to address the highly correlated noise present in JPEG images.
- Analysis of research results is not in-depth. The analysis overemphasized the importance of the results and did not explain the value of the research in depth.
Besides the discussion of the results already included in the first version of the manuscript, we updated the last section ("Conclusions, limitations and future research"). In this section, we included a detailed analysis of the implications of our research as well as its limitations and ideas for future research.
Reviewer 2 Report
Report on the manuscript "jimaging-1263231" entitled "Forgery Detection in Digital Images by Multi-Scale Noise Estimation"
This manuscript proposes a multi-scale approach for detecting image forgery by analyzing highly correlated noise identified in digital images. The authors estimate a noise curve for each image block, in each color channel, and at each scale. Some experimental results are presented to illustrate the proposed approach. A discussion and conclusions about the present investigation are reported.
In general, I have a good opinion about this work and recommend its acceptance after a minor revision that considers the following concerns:
1. The manuscript needs to be proofread by the authors carefully. I recommend the authors ask for the assistance of an English native speaker.
2. Words in the title are not usually in the keywords. In addition, the keywords are often written in alphabetical order.
3. To the best of my knowledge, there has been a good body of work done in the literature on this topic. The bibliographical review might be improved.
4. The authors must check the use of all acronyms and abbreviations employed in the whole manuscript. Although known, for a general audience that could be interested in your work, please define "JPEG" and present it in the keywords.
5. Notations and mathematical symbols must be checked and fixed. Please remove the use of "\cdot" (in Latex code) for expressing a multiplication (usual mathematical product); see page 7 but check it in the whole manuscript. This can be confused with other unusual mathematical products.
6. Fractions used in a line of text must be written with "/" instead of using "\frac" (in Latex code); see, for example, line 153.
7. Please do not use Italic style for any text in the whole manuscript, especially for acronyms in formulas, as "MCC, IoU, etcetera". Please keep in mind that the Italic font is used for the Latin language (as for example "et al.") and for mathematical variables.
8. In my opinion, the implications of the study are underdeveloped and must be explained further in the sections of discussion/conclusions.
9. The authors must provide more details about the computational framework used in the manuscript. For example, software and packages used, features of the computer employed, runtimes, and other computational aspects must be added.
10. I recommend summarizing the methodology in a pseudo-algorithm (only based on text with no notations) obtained from Algorithm 1 so that the readers can follow it easier.
11. I do not have checked each formula or numerical results in detail. I recommend the authors to check them.
12. The final section ("Discussion") needs to be improved. The authors must add more conclusions, limitations of the study, and ideas for further research. Then, I suggest titling the final section as "Conclusions, limitations, and future research".
13. The authors must check whether all references are cited and whether all citations are in the reference list, as well as making an effort to discuss and cite any papers on the topic published in the Journal of Imaging to attract the attention of our target audience.
Author Response
We would like to thank you for taking the time to assess our manuscript. We address your suggestions below and in the updated version of the manuscript. Major changes in the manuscript are highlighted in red.
- The manuscript needs to be proofread by the authors carefully. I recommend the authors ask for the assistance of an English native speaker.
We carefully proofread the revised version of the manuscript.
- Words in the title are not usually in the keywords. In addition, the keywords are often written in alphabetical order.
Done.
- To the best of my knowledge, there has been a good body of work done in the literature on this topic. The bibliographical review might be improved.
We have updated the bibliographical review in order to also include deep-learning approaches, besides the ones mentioned in the first version.
- The authors must check the use of all acronyms and abbreviations employed in the whole manuscript. Although known, for a general audience that could be interested in your work, please define "JPEG" and present it in the keywords.
We added an abbreviations list to the manuscript where all the abbreviations used in the article are explained. Furthermore, we added "JPEG" to the keywords.
- Notations and mathematical symbols must be checked and fixed. Please remove the use of "\cdot" (in Latex code) for expressing a multiplication (usual mathematical product); see page 7 but check it in the whole manuscript. This can be confused with other unusual mathematical products.
Done.
- Fractions used in a line of text must be written with "/" instead of using "\frac" (in Latex code); see, for example, line 153.
Done.
- Please do not use Italic style for any text in the whole manuscript, especially for acronyms in formulas, as "MCC, IoU, etcetera". Please keep in mind that the Italic font is used for the Latin language (as for example "et al.") and for mathematical variables.
Done.
- In my opinion, the implications of the study are underdeveloped and must be explained further in the sections of discussion/conclusions.
Besides the discussion of the results already included in the first version of the manuscript, we updated the last section ("Conclusions, limitations and future research"). In this section, we included a detailed analysis of the implications of our research.
- The authors must provide more details about the computational framework used in the manuscript. For example, software and packages used, features of the computer employed, runtimes, and other computational aspects must be added.
We added an appendix including the implementation details requested.
- I recommend summarizing the methodology in a pseudo-algorithm (only based on text with no notations) obtained from Algorithm 1 so that the readers can follow it easier.
Done.
- I do not have checked each formula or numerical results in detail. I recommend the authors to check them.
We have checked all the formulas and results presented in the manuscript. We did not find any errors.
- The final section ("Discussion") needs to be improved. The authors must add more conclusions, limitations of the study, and ideas for further research. Then, I suggest titling the final section as "Conclusions, limitations, and future research".
We updated the last section (and renamed it "Conclusions, limitations and future research"). In this section, we included a detailed analysis of the implications of our research as well as its limitations of the method and ideas for future research.
- The authors must check whether all references are cited and whether all citations are in the reference list, as well as making an effort to discuss and cite any papers on the topic published in the Journal of Imaging to attract the attention of our target audience.
We checked all references and added new ones, including [16] and [13] from the Journal of Imaging.
Round 2
Reviewer 1 Report
Authors anwsered my comments.